# Treatment and Outcomes of *Clostridioides difficile* Infection in Switzerland: A Two-Center Retrospective Cohort Study

**DOI:** 10.3390/jcm11133805

**Published:** 2022-06-30

**Authors:** Paraskevas Filippidis, Eleftheria Kampouri, Maximilian Woelfle, Tina Badinski, Antony Croxatto, Tatiana Galperine, Matthaios Papadimitriou-Olivgeris, Bruno Grandbastien, Yvonne Achermann, Benoit Guery

**Affiliations:** 1Infectious Diseases Service, Lausanne University Hospital, University of Lausanne, Rue du Bugnon 46, 1011 Lausanne, Switzerland; paraskevas.filippidis@chuv.ch (P.F.); kampourie@fredhutch.org (E.K.); katerina-tatiana.galperine@chuv.ch (T.G.); matthaios.papadimitriou-olivgeris@chuv.ch (M.P.-O.); 2Service of Hospital Preventive Medicine, Lausanne University Hospital, University of Lausanne, Mont Paisible 18, 1011 Lausanne, Switzerland; bruno.grandbastien@chuv.ch; 3Division of Infectious Diseases and Hospital Epidemiology, University Hospital Zurich, Rämistrasse 100, 8091 Zurich, Switzerland; maximilian.woelfle@gmail.com (M.W.); tina.badinski@uzh.ch (T.B.); yvonne.achermann@usz.ch (Y.A.); 4Institute of Microbiology, Department of Medical Laboratory and Pathology, University Hospital, University of Lausanne, Rue du Bugnon 48, 1011 Lausanne, Switzerland; antony.croxatto@ne.ch; 5Internal Medicine, Hospital Zollikerberg, Trichtenhauserstrasse 20, 8125 Zollikerberg, Switzerland

**Keywords:** *Clostridioides difficile*, outcomes, recurrence, severe infection, mortality, treatment, predictive factors

## Abstract

**Objectives**: *Clostridioides difficile* infection (CDI) is the leading cause of healthcare-associated diarrhea, often complicated by severe infection and recurrence with increased morbidity and mortality. Data from large cohorts in Switzerland are scarce. We aimed to describe diagnostic assays, treatment, outcomes, and risk factors for CDI in a large cohort of patients in Switzerland. **Methods**: We conducted a retrospective cohort study of CDI episodes diagnosed in patients from two tertiary care hospitals in Switzerland. During a 3-month follow-up, we used a composite outcome combining clinical cure at day 10, recurrence at week 8, or death, to evaluate a patient’s response. Unfavorable outcomes consisted in the occurrence of any of these events. **Results**: From January 2014 to December 2018, we included 826 hospitalized patients with documented CDI. Overall, 299 patients (36.2%) had a severe infection. Metronidazole was used in 566 patients (83.7%), compared to 82 patients (12.1%) treated with vancomycin and 28 patients (4.1%) treated with fidaxomicin. Overall mortality at week 8 was at 15.3% (112/733). Eighty-six patients (12.7%) presented with clinical failure at day 10, and 78 (14.9%) presented with recurrence within 8 weeks; 269 (39.8%) met the composite outcome of death, clinical failure, or recurrence. The Charlson Comorbidity Index score (*p* < 0.001), leukocytes > 15 G/L (*p* = 0.008), and the use of metronidazole (*p* = 0.012) or vancomycin (*p* = 0.049) were factors associated with the composite outcome. **Conclusions**: Our study provides valuable insights on CDI treatment and outcomes in Switzerland, highlights the heterogeneity in practices among centers, and underlines the need for the active monitoring of clinical practices and their impact on clinical outcomes through large multicentric cohorts.

## 1. Introduction

*Clostridioides* (formerly *Clostridium*) *difficile* is the leading cause of healthcare-associated diarrhea, responsible for 12% of all healthcare-associated infections (HAI) in the U.S. [1] and 4.3% of all HAI in Switzerland [2]. Of note, the reported incidence of CDI is influenced by the diagnostic methods used [3,4,5]. Diagnostic approaches have considerably evolved over the years [6,7]; the introduction and wide use of nucleic acid amplification tests (NAAT) offers great advantages, including high sensitivity and short turnaround times, but has raised concerns about the overdiagnosis of CDI [3,4,5]. In a nationwide multicenter point-prevalence study in Switzerland, the mean prevalence rate was 9.3 cases/10,000 patient bed-days when using a PCR detection method, and lower when using a two-stage enzyme immunoassay (EIA)-based algorithm (6.1 cases/10,000 patient bed-days) [8]. Despite the advancements in diagnostic methods, the optimal diagnostic algorithm remains debated, and strategies vary considerably between centers. A better appreciation of local diagnostic approaches is vital for evaluating CDI burden [8,9].

The treatment of CDI has also substantially evolved during the last decade; the use of metronidazole has largely been abandoned in favor of vancomycin and fidaxomicin for severe and non-severe first episodes as well as recurrences. The 2017 guidelines from the Infectious Diseases Society of America and the Society of Healthcare Epidemiology of America (IDSA/SHEA) issued a strong recommendation for the use of either molecule over metronidazole due to their superiority in clinical cure and recurrences [7,10,11]. More recently, in view of the clear-cut advantage of fidaxomicin over vancomycin on sustained clinical response and recurrences and its favorable safety profile [12,13,14,15], both the 2021 IDSA/SHEA and the 2021 guidelines by the European Society of Clinical Microbiology and Infectious Diseases (ESCMID) marked the shift from vancomycin to fidaxomicin from the first episode [16,17]. Despite concerns regarding its increased cost, fidaxomicin was considered cost-effective and easy to implement. Vancomycin is still considered an acceptable alternative, though it is important to stress that existing resources and logistical issues influence the implementation of guidelines [16].

Given the rapidly transforming context in CDI treatment and the ongoing challenges in treatment allocation, there is an unmet need for local data on treatment practices and outcome assessment to evaluate such practices. In a recent epidemiologic study conducted in two tertiary Swiss hospitals between 2014 and 2018, we reported the clinical characteristics of 826 patients with microbiologically documented CDI [18]. The aim of the present study was to describe the diagnostic methods, treatments, and clinical outcomes and to identify the risk factors correlated with worse outcomes.

## 2. Methods

We conducted a retrospective study of all consecutive CDI cases that occurred in adult patients hospitalized in two tertiary care centers: the Lausanne University Hospital and the University Hospital of Zürich, between 2014 and 2018. Suspected cases of CDI were identified from the microbiology laboratory database based on a positive test and were confirmed by a records review. Patients were included if they had received a CDI diagnosis (meeting the IDSA/SHEA and ESCMID criteria of diarrhea and a positive CDI test) [7,17,19]. The tests that were used were combinations of glutamate dehydrogenase (GDH) (Techlab, Blacksburg, VA, USA), a polymerase chain reaction (PCR) test for toxigenic *C. difficile* (GeneXpert *C. difficile*, Cepheid, Sunnyvale, CA, USA), and enzyme immunoassays (EIA) (Techlab, Blacksburg, VA, USA) for toxins depending on the center.

We collected demographic, clinical, and biological data from a review of medical records. We specifically collected data on factors associated with severe disease and/or recurrence, such as age (dichotomized as age above or below 65 years), Charlson Comorbidity Index (CCI) score [20], previous (during the last 3 months) or current antibiotic use or anti-acid use, active malignancy, and current chemotherapy. We evaluated severity criteria as defined by the IDSA/SHEA guidelines: an elevated serum creatinine level of >1.5 mg/dL (133 mcmol/L) and/or marked leukocytosis (≥15 G/L) [7]. The date of diagnosis was defined as day 0 (D0). We collected data on D0 when available or the ‘worse value’ available in the 48-h interval around the time of diagnosis. Treatment regimen, dosing, and duration were at the discretion of the treating physician. The primary endpoint was a composite of clinical failure and/or recurrence and/or mortality. Clinical failure was defined as the absence of a clinical response (improvement of symptoms) by day 10. Recurrence was defined as a new onset of diarrhea or other compatible clinical presentation occurring after an initial improvement upon treatment, with a new positive test and requiring of treatment, within 8 weeks from the initial episode. Mortality was evaluated at week 8. Severe disease was defined as per the IDSA criteria. The study was approved by the local ethics committees (ID 2018-01330; KEK Zurich Nr. 2016-00145).

### Data Collection and Statistical Analysis

Data were collected using REDCap electronic data capture tools [21,22]. REDCap (Research Electronic Data Capture) is a secure web-based software platform designed to support data capture for research studies. Descriptive statistics (mean and standard deviation (SD), median and interquartile range (IQR), and percentage) were reported for key variables. The Wilcoxon rank sum test, Pearson’s chi-squared test, and Fisher’s exact test were performed as appropriate. All variables with *p* < 0.2 in the univariate analysis were included in a multivariate model. The results are presented as odds ratios (ORs) with their 95% confidence intervals (95% CIs) calculated using the generalization of the ordinary linear regression method. A value of *p* < 0.05 was considered statistically significant. Variance inflation factor (VIF) was calculated to quantify multicollinearity. All analyses were performed using the R software version 3.6.0 (2019) (R Foundation for Statistical Computing, Vienna, Austria; URL https://www.R-project.org/, accessed on 26 June 2022).

## 3. Results

From January 2014 to December 2018, we identified 923 hospitalized patients with microbiologically documented CDI. We included 826 patients with first episodes and available clinical data who had previously provided signed general consent (Figure 1). The median age was 67 years (IQR: 54–78) and the median CCI score was five (IQR: 3–7). A total of 649 patients (81.4%) had received antibiotics in the last 3 months and 153 patients (18.5%) had active malignancies and were receiving concomitant chemotherapy upon CDI diagnosis. Overall, 299 patients (36.2%) had a severe infection based on the IDSA criteria.

For the diagnosis of CDI, the presence of a positive glutamate dehydrogenase (GDH) test was systematically assessed in both centers throughout the study period, with 795 positive tests (96.2%). Overall, nucleic acid amplification tests (NAATs) were positive in 537 patients (65.0%), ranging from 5 (4.4%) in 2014 to 147 (99.3%) in 2018 (Table 1); the increasing rate of positive NAATs over the years (from 4.4% in 2014 to 100% in 2018) reflects the change in testing practices in Lausanne with an increase in the use of NAATs in recent years (Appendix A). The enzyme immunoassay (EIA) for toxin A/B was positive in 256 patients (31.0%), ranging from 46 (40.7%) positive tests in 2014 to 42 positive tests (28.4%) in 2018. Of note, EIA testing for toxin A/B was performed only in Lausanne during the study period (Appendix A). Toxigenic culture and the detection of binary toxin or ribotype O27 strains were only performed in Lausanne (Appendix A). The diagnostic tests and positivity rates per study year in both centers are shown as pooled data in Table 1.

Metronidazole was the most common treatment in both centers and was used in 563 patients overall (83.3%, 563/676); 86 patients (12.7%, 86/676) received vancomycin and 27 patients (4.0%, 27/676) fidaxomicin. Fidaxomicin was exclusively prescribed in Lausanne. Updated IDSA/SHEA guidelines on CDI management were issued in April 2017 and could have impacted standard practice. The use of vancomycin or fidaxomicin was favored over that of metronidazole in Lausanne; metronidazole was used in up to 90% of patients before 2018 and decreased to 40% after 2018. No such change in practice was observed in Zurich (Appendix A).

All-cause mortality at week 8 was estimated at 15.3% (112/733), with 50.9% of these patients (57/112) dying within 10 days from diagnosis. Eighty-six patients (12.7%; 86/676) presented clinical failure at day 10, and 14.9% (78/522) presented a recurrence within 8 weeks (Figure 1). Seven patients died after clinical failure.

We performed univariate and multivariate analyses for the composite outcome of death at day 10 and week 8, clinical failure at day 10, or recurrence at week 8 (Table 2 and Table 3). The analysis included 676 patients for whom a complete outcome evaluation was available. Among these 676 patients, 269 presented the composite outcome. The univariate analysis found an age of ≥65 years (*p* < 0.001), CCI score (*p* < 0.001), leukocytes > 15 G/L (*p* = 0.022), creatinine > 132 µmol/L (*p* = 0.001), treatment other than fidaxomicin (*p* = 0.026), and severe disease based on the IDSA criteria (*p* < 0.001) to be associated with the composite outcome. Active malignancy and concomitant chemotherapy were not associated with the clinical outcome (*p* > 0.9 and *p* = 0.3, respectively) (Table 2). The multivariate logistic regression model identified CCI score (OR 1.11, 95% CI: 1.04–1.18; *p* < 0.001), leukocytes > 15 G/L (OR 1.77, 95% CI: 1.16–2.69; *p* = 0.008), and the use of metronidazole (OR 4.10, 95% CI: 1.49–14.4; *p* = 0.012) or vancomycin (OR 3.29, 95% CI: 1.09–12.4; *p* = 0.049) as predictive factors independently associated with the composite outcome. Multivariate analyses for all outcomes are presented in Table 3. No significant difference was found between metronidazole and vancomycin for the different outcomes (Appendix A).

## 4. Discussion

In a large cohort of 826 inpatients from two tertiary hospitals, we showed that CDI is frequently associated with unfavorable outcomes, including recurrence, clinical failure (day 10), and/or all-cause mortality, which were present in almost 40% of patients. Both diagnostic and treatment practices varied substantially between the two centers. Metronidazole was largely prescribed in both centers and both metronidazole and vancomycin were strongly associated with worse outcomes when compared to fidaxomicin. Finally, marked leukocytosis (WBC > 15 G/L) and the CCI score were identified as being predictive factors for the composite outcome of clinical failure, recurrence, and/or death.

Consistent with global trends and current diagnostic recommendations [6], the percentage of cases diagnosed by means of NAAT increased from 4% in 2014 and 54% in 2015 to a peak of 99% in 2018. The diagnostic algorithms used varied substantially between the two centers: all CDI diagnoses in Zurich were based solely on GDH in combination with NAAT (without EIA toxin detection), whereas EIA toxin detection was present in 61% of cases in Lausanne. Several studies have shown that toxin-positive CDIs are more severe with worse outcomes, and that the reliance on NAAT diagnostic testing might lead to substantial overdiagnosis [23,24,25]. This difference in diagnostic algorithms might have led to an overestimation of ‘true’ infections in Zurich and the inclusion of more severe cases in Lausanne, which could have influenced outcomes. However, our results reflect real life data since our case definition required diagnosis and CDI treatment by the treating physician, and the combination of positive NAAT and GDH is sufficient for the diagnosis of CDI in the appropriate clinical setting. Our findings underline the heterogeneity in diagnostic methodology and the resulting difficulties in accurately estimating CDI burden.

In our study, metronidazole was largely prescribed (83.7%). This finding agrees with another Swiss study of 210 CDI episodes between 2014 and 2016, where metronidazole was used in 94% of first episodes and excessively frequently in recurrences (between 50–70% depending on the number of recurrences). Of note, all patients in the previous study and the majority of patients in our study were treated before the updated 2017 IDSA/SHEA guidelines [7]. In our study, the use of metronidazole subsequently declined only in Lausanne in 2018 and remained unchanged in Zurich. Approximately 40% of patients treated with metronidazole or vancomycin presented with unfavorable outcomes. Both metronidazole (OR= 4.10, 95% CI: 1.49–14.4; *p* = 0.012) and vancomycin (OR = 3.29, 95% CI: 1.09–12.4; *p* = 0.049) were associated with an increased risk for the composite outcome. In our study, both molecules seemed to perform comparably (Appendix A). The lack of superiority of vancomycin over metronidazole could be explained by the higher comorbidity burden seen in patients treated with vancomycin when compared to those treated with metronidazole. However, the lack of stratification according to the severity of CDI episodes precludes an accurate comparison between these molecules.

The superiority of fidaxomicin for sustained clinical cure and recurrences has been established by large RCTs [12,13,14,15], and these findings have prompted a recommendation shift from vancomycin to fidaxomicin from the first, non-severe episode in the 2021 IDSA/SHEA and ESCMID guidelines [16,17]. The pooled analysis of these studies confirmed that fidaxomicin had an increased sustained response at 4 weeks after the end of therapy but did not reduce all-cause mortality. Furthermore, both initial clinical cure and sustained response at 90 days were comparable between fidaxomicin and vancomycin [12,13,14,15,16]. Our study, examining a composite outcome of clinical failure, recurrence at week 8 (as per IDSA definition), and/or mortality, demonstrated a protective effect of fidaxomicin. However, this effect was not observed when examining recurrences compared to metronidazole (OR = 1.77, 95% CI: 0.46–11.6) or vancomycin (OR = 1.01, 95% CI: 0.20–7.46), probably because our study subgroups were underpowered to detect such a difference. The changing landscape of CDI treatment due to the advent of more efficient and safer molecules calls for a more thorough assessment of current treatment practices and of the associated clinical outcomes.

The all-cause mortality rate of approximately 15% of the patients observed in our study is within the range frequently reported in the literature in hospitalized patients outside outbreaks [26,27,28]. Our findings are consistent with another Swiss study reporting a rate of all-cause mortality at 60 days of 16% [29], as well as one large Dutch study among 1366 patients reporting a rate of 13% after 30 days [27]. We report a recurrence rate of 15%, which is within the range of 10–30% usually reported in the literature [30,31], and lower than what has been previously described in Switzerland (20%) [29]. When comparing the outcomes between the two centers, we observed a lower rate of clinical failure in the Zurich center compared to the Lausanne center (OR = 5.35, 95% CI: 2.92–10.4; *p* < 0.001), which probably reflects the differences in treated populations and diagnostic algorithms in the two centers. The Lausanne participants were older (median 70 years, *p* < 0.001) and had a higher comorbidity burden (median CCI score at 5.5, *p* < 0.001) and more frequent severe disease (up to 40%, *p* = 0.03). Interestingly, the recurrence rate was still higher in Zurich (OR = 1.97, 95% CI: 1.13–3.60; *p* = 0.016), which could be explained, at least to some extent, by the most common use of metronidazole in this center.

The severity of disease is still one of the main determinants when choosing treatment and one of the most confusing terms in the CDI literature, with no consensus regarding definition among guidelines [7,19,32,33]. In general, severe disease refers to the presence of prognostic factors associated with unfavorable outcomes (complicated/fulminant disease, treatment failure, ICU admission, or death). The ESCMID guidelines propose a wide variety of prognostic markers, among which four are classified with a strong recommendation: advanced age (>65 years old), leukocytosis, low albumin levels, and increased creatinine levels [19]. To simplify, the IDSA/SHEA guidelines define severe disease by the presence of marked leukocytosis (>15 G/L) and/or a serum creatinine level of >1.5 mg/dl and stress the need for validation in large cohorts [7]. In our study, marked leukocytosis and the CCI score were predictors of the composite outcome of clinical failure, recurrence, and/or death. Of note, the CCI score was the factor with the strongest statistically significant association with composite and mortality outcomes, in both univariate and multivariate models (Table 3).

Our findings are supported by a recent systematic review of 31 studies [33]. Leukocyte count, albumin, creatinine, age, comorbidities, and ICU admission were the most common identified risk factors of severity. However, when the authors differentiated studies according to methodology and quality of study, they identified age and the presence of comorbidities as the two most common severity predictors for CDI. CCI score was frequently used to assess comorbidity burden in several of the included studies [34,35,36,37,38]. Interestingly, even though the role of leukocytosis in severity classification is widely recognized, age and comorbidities (as depicted by validated scores such as the CCI score) are not included in any of the currently available international guidelines to define severe disease. Whether those two additional factors should be considered in the definition of severity and, consequently, in treatment allocation, must be validated in large, prospective cohorts.

Our study has several strengths. First, this is the biggest cohort in Switzerland to date, with data from two different tertiary care centers, representative of the different practices in different regions of Switzerland. This is of value given the absence of routine surveillance for CDI on a national level and the scarcity of epidemiological data in our country. Second, this study is timely since it provides an overview of treatment practices before the wider implementation of the recently published guidelines by international societies. Finally, in our study, we used robust definitions of outcomes and all patients were diagnosed and followed by physicians, thus providing a reliable evaluation of outcomes.

Several limitations of our study are also worth noting. First, this a retrospective study with substantial differences in diagnosis and management between participating centers, which might have impacted the pooling of data. NAAT was used in 64% of all patients, and this, along with the infrequent toxin detection by EIA, could be responsible for some degree of overdiagnosis and the inclusion of ‘less severe’ infections in the later years. Second, CDI treatments were not stratified for disease severity or comorbidity burden; thus, we were not able to provide a reliable explanation of metronidazole’s non-inferiority in terms of the recurrence rate when compared to vancomycin. Most importantly, although both metronidazole and vancomycin were associated with worse outcomes, the fidaxomicin group was clearly underpowered, limiting our ability to safely evaluate its efficacy. Finally, the time of follow-up of 8 weeks, though sufficient to capture recurrences and longer than what is used in most RCTs, hinders a better understanding of long-term outcomes.

## 5. Conclusions

Our study provides valuable insights into CDI treatment practices and associated outcomes in Switzerland, where epidemiological data are needed. Our findings highlight the heterogeneity in both diagnosis and management among different centers, even within the same country. As we move forward towards the wider implementation of newer treatments, there is an imperative need for large multicentric cohorts to identify risk factors for worse outcomes, assess treatment practices and associated outcomes, and ensure long-term follow-up. This study is the first step towards that direction in Switzerland.

## Figures and Tables

**Figure 1 jcm-11-03805-f001:**
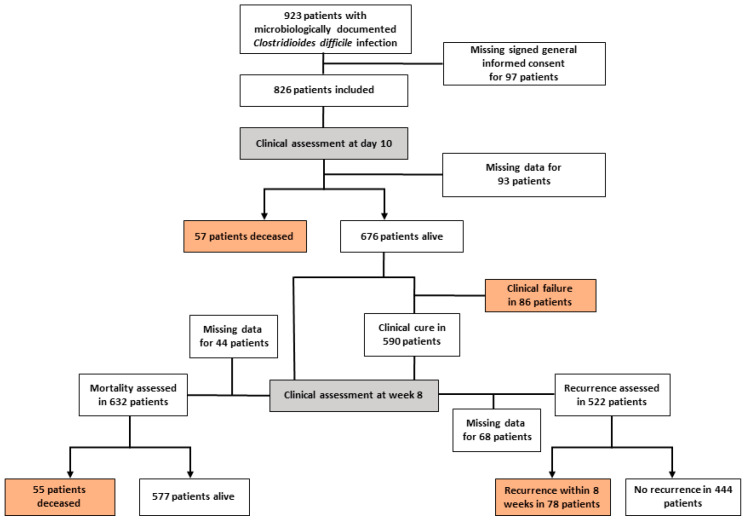
Flowchart.

**Table 1 jcm-11-03805-t001:** Evolution of diagnostic tests over time.

Characteristic	2014 N = 113 ^1^	2015 N = 165 ^1^	2016 N = 162 ^1^	2017 N = 238 ^1^	2018 N = 148 ^1^	Overall, N = 826 ^1^
NAAT	4.4% (5)	55.8% (92)	87.0% (141)	63.9% (152)	99.3% (147)	65.0% (537)
GDH	95.6% (108)	99.4% (164)	99.4% (161)	92.0% (219)	96.6% (143)	96.2% (795)
Toxin A/B EIA	40.7% (46)	38.8% (64)	29.6% (48)	23.5% (56)	28.4% (42)	31.0% (256)
Toxigenic culture	32.7% (37)	0.0% (0)	4.3% (7)	18.1% (43)	0.0% (0)	10.5% (87)
Ribotype 027	0.0% (0)	0.0% (0)	0.0% (0)	0.8% (2)	0.0% (0)	0.2% (2)
Binary toxin positive	0.0% (0)	0.0% (0)	0.0% (0)	5.5% (13)	3.4% (5)	2.2% (18)

^1^ % [n].

**Table 2 jcm-11-03805-t002:** Univariate analysis of factors associated with composite outcome.

Characteristic	Outcome Not Met N = 407	Outcome Met N = 269	*p*-Value
** *Demographics* **			
Age	65 (52, 75)	71 (58, 80)	<0.001
Age group			<0.001
<65	68.0% (200)	32.0% (94)	
≥65	54.2% (207)	45.8% (175)	
Sex			0.3
Female	62.1% (192)	37.9% (117)	
Male	58.6% (215)	41.4% (152)	
** *Comorbidities* **			
Charlson comorbidity score	5.0 (3.0, 7.0)	6.0 (4.0, 8.0)	<0.001
Active malignancy	60.0% (105)	40.0% (70)	>0.9
** *Laboratory findings* **			
Leukocytes (G/L)	8 (5, 13)	10 (6, 15)	0.040
Leukocytes classes			0.022
<0.5 G/L	64.1% (25)	35.9% (14)	
0.5–15 G/L	64.9% (289)	35.1% (156)	
>15 G/L	51.6% (66)	48.4% (62)	
Creatinine > 132 µmol/L	49.4% (87)	50.6% (89)	0.001
** *Treatments* **			
CDI treatment			0.026
Fidaxomicin	85.2% (23)	14.8% (4)	
Metronidazole	59.1% (333)	40.9% (230)	
Vancomycin	59.3% (51)	40.7% (35)	
Antiacid treatment			0.2
No	55.6% (124)	44.4% (99)	
Yes	62.5% (277)	37.5% (166)	
Antibiotic treatment (≤3 mois)			>0.9
No	61.2% (71)	38.8% (45)	
Yes	60.1% (328)	39.9% (218)	
Concomitant chemotherapy	63.8% (90)	36.2% (51)	0.3
***Clinical scores*** IDSA severity group			<0.001
Non severe	65.3% (271)	34.7% (144)	
Severe	52.1% (136)	47.9% (125)	

**Table 3 jcm-11-03805-t003:** Multivariate analyses for the composite outcome ^1^, death at day 10, death at week 8, clinical failure and recurrence at week 8.

	Composite Outcome		Death at Day 10 (N = 57)		Death at Week 8 (N = 112)		Clinical Failure (N = 86)		Recurrence at Week 8 (N = 78)	
Characteristic	OR (95% CI) ^2^	*p*-Value	OR (95% CI) ^2^	*p*-Value	OR (95% CI) ^2^	*p*-Value	OR (95% CI) ^2^	*p*-Value	OR (95% CI) ^2^	*p*-Value
Age group				**0.001**				0.11		
<65			1.00				1.00			
≥65			4.26 (1.69, 12.6)	0.004			1.60 (0.91, 2.91)			
Charlson comorbidity score	1.11 (1.04, 1.18)	**<0.001**	1.25 (1.10, 1.41)	**<0.001**	1.26 (1.14, 1.40)	**<0.001**				
Leukocytes classes		**0.022**		**0.008**						
0.5–15 G/L	1.00		1.00							
<0.5 G/L	1.44 (0.69, 2.90)	0.3	6.18 (1.57, 20.9)	0.005						
>15 G/L	1.77 (1.16, 2.69)	0.008	2.03 (1.05, 3.85)	0.031						
Creatinine	1.40 (0.94, 2.08)	0.093							1.69 (0.98, 2.88)	0.059
CDI treatment		**0.016**		**0.019**						0.4
Fidaxomicin	1.00								1.00	
Metronidazole	4.10 (1.49, 14.4)	0.012	1.00						1.77 (0.46, 11.6)	
Vancomycin	3.29 (1.09, 12.4)	0.049	1.90 (0.86, 3.97)	0.10					1.01 (0.20, 7.46)	
Center								**<0.001**		**0.017**
Lausanne							1.00		1.00	
Zurich							0.19 (0.10, 0.34)	<0.001	1.97 (1.13, 3.57)	0.021

^1^ Composite outcome = Death at day 10, death at week 8, clinical failure at day 10 and/or recurrence at week 8. ^2^ OR = Odds Ratio, CI = Confidence Interval.

## Data Availability

The datasets used during the current study are available from the corresponding author on reasonable request.

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
