# Peer review of "Treatment and Outcomes of Clostridioides difficile Infection in Switzerland: A Two-Center Retrospective Cohort Study"

_jcm, 2022, doi:10.3390/jcm11133805_

Round 1
Reviewer 1 Report
This paper - another non RCT and inappropriate conclusions. But I think it is an outstanding paper with fascinating data and very well referenced. Did anybody in the cohort get metronidazole as a first treatment and fidaxomicin if they failed or for the recurrence?
Author Response
Dear colleague, we thank you for your very nice review of our work. To answer your question, a total of 7 patients received fidaxomicin after metronidazole failure. One treatment was administered for a first recurrence and the additional 6 were for multiple recurrence. For the first recurrence vancomycin was the drug chosen when fidaxomicin was not. Finally, fidaxomicin was prescribed only in Lausanne and not in Zurich.
Best regards
Benoit Guery
Reviewer 2 Report
This paper described retrospective cohort study of CDI epizodes diagnosed in patients from 2 tertiary care hospitals in Switzerland. During 3 months follow- up Authors used a composite outcome combining clinical cure at day 10, reccurence at week 8, or death, to evaluate patient's response. Study was conducted between January 2014 and december 2018. Authors concluded that this study provides valuable insights on CDI treatment and outcomes in Switzerland and highlights the heterogeneity in practicies among centers and underlines the need for active monitoring of clinical practices and their impact on clinical outcomes through large multicentric cohorts.
This study is very important, because describes diagnosis, treatment and also complications among CDI patients. metodology and statistics used is also acceptable. So, I recommend this paper for publication. I recommend to include in the methods section used diagnostic tests'names (Co, country).
Author Response
Dear colleague,
We thank you for your review, as suggested we added the diagnostic test names. Extensive editing of the english will be performed through the MDPI platform on the revised manuscript as suggested.
Best regards
Benoit Guery